# Critical research gaps in treating growth faltering in infants under 6 months: A systematic review and meta-analysis

Cecília Tomori[1,2]*, Deborah L. O'Connor[3], Mija Ververs[4], Dania Orta-Aleman[4], Katerina Paone[5], Chakra Budhathoki[1], Rafael Pérez-Escamilla[5]

**1** Johns Hopkins University School of Nursing, Johns Hopkins University, Baltimore, Maryland, United States of America, **2** Department of Population, Johns Hopkins University Bloomberg School of Public Health, Family and Reproductive Health, Johns Hopkins University, Baltimore, Maryland, United States of America, **3** Temerty Faculty of Medicine, Department of Nutritional Sciences, University of Toronto, Toronto, Ontario, Canada, **4** Department of International Health, Johns Hopkins University Bloomberg School of Public Health, Johns Hopkins University, Baltimore, Maryland, United States of America, **5** Department of Social and Behavioral Health, Yale University School of Public Health, Yale University, New Haven, Connecticut, United States of America

* ctomori1@jhu.edu

**Data Availability Statement:** Data for this systematic review are presented in the manuscript and supporting documentation.

## Abstract

In 2020, 149.2 million children worldwide under 5 years suffered from stunting, and 45.4 million experienced wasting. Many infants are born already stunted, while others are at high risk for growth faltering early after birth. Growth faltering is linked to transgenerational impacts of poverty and marginalization. Few interventions address growth faltering in infants under 6 months, despite a likely increasing prevalence due to the negative global economic impacts of the COVID-19 pandemic. Breastfeeding is a critical intervention to alleviate malnutrition and improve child health outcomes, but rarely receives adequate attention in growth faltering interventions. A systematic review and meta-analysis were undertaken to identify and evaluate interventions addressing growth faltering among infants under 6 months that employed supplemental milks. The review was carried out following guidelines from the USA National Academy of Medicine. A total of 10,405 references were identified, and after deduplication 7390 studies were screened for eligibility. Of these, 227 were assessed for full text eligibility and relevance. Two randomized controlled trials were ultimately included, which differed in inclusion criteria and methodology and had few shared outcomes. Both studies had small sample sizes, high attrition and high risk of bias. A Bangladeshi study (n = 153) found significantly higher rates of weight gain for F-100 and diluted F-100 (DF-100) compared with infant formula (IF), while a DRC trial (n = 146) did not find statistically significant differences in rate of weight gain for DF-100 compared with IF offered in the context of broader lactation and relactation support. The meta-analysis of rate of weight gain showed no statistical difference and some evidence of moderate heterogeneity. Few interventions address growth faltering among infants under 6 months. These studies have limited generalizability and have not comprehensively supported lactation. Greater investment is necessary to accelerate research that addresses

**Funding:** This work was supported by the World Health Organization (WHO) (APW 202859904 to CT). WHO commissioned the review and as such provided guidance on the development of the questions examined by the review, and reviewed and provided methodological feedback on the technical report. WHO did not review or provided feedback on the interpretation of findings or recommendations from this article. These were made by the article's authors and do not necessarily represent the views from the WHO.

**Competing interests:** The authors have declared that no competing interests exist.

growth faltering following a new research framework that calls for comprehensive lactation support.

## Introduction

A substantial portion of infants face growth faltering each year across the globe [1–4]. While definitions vary, for the purpose of this paper the term growth faltering refers to a slower rate of weight, length or head circumference gain than expected for a children's age and sex [5]. Of the 149 million children globally under 5 years who are stunted (length-for-age z score less than 2 standard deviation scores below the World Health Organization (WHO) child growth standards median [6]), recent research indicates that 15% are born stunted, and an additional 25% become stunted during the first 6 months after birth [7]. Additionally, 45.4 million children under 5 years of age, constituting approximately 6.7% of children, experienced wasting (weight-for-length z score less than 2 standard deviation scores below the WHO child growth standards median [6]) [4]. Infants who experience growth faltering and do not receive adequate nutritional interventions are at increased risk for mortality and morbidity from infectious illness and from non-communicable diseases [8–11]. The causes of growth faltering are complex and multifactorial. Preterm births (PTB), and infants born with low birth weight (LBW, <2500g), and small for gestational age (SGA, birth weight <10th percentile) constitute a substantial portion of those at risk across the globe for growth faltering [6,12]. Multiple comorbidities may increase the risk of growth faltering.

Importantly, social determinants of health (SDOH), such as poverty and marginalization, are key drivers of conditions that put infants at risk for growth faltering. For instance, mothers with short stature experience higher rates of LBW and preterm births [13]. Short stature itself is the outcome of intergenerational poverty [14]. Impoverished women are at higher risk for infectious diseases such as malaria and HIV, both causes of PTB, LBW and SGA [15–17]. COVID-19 is also associated with a higher risk of preterm birth [18]–and with inequities. Poverty also drives exposures to infectious agents in the home environment due to lack of access to proper sanitation including clean water, toilets, clean housing among other factors [19,20]. Poor and marginalized women across settings are at high risk for a number of complications during pregnancy [21], including diabetes and hypertension, which are associated with a substantial burden of adverse birth outcomes [22,23]. Importantly, as the food environment is rapidly changing in many settings, people in LMIC are often facing a double burden of malnutrition, with rising childhood obesity and increasing stunting at the same time [24,25]. Additional environmental drivers of PTB, LBW and SGA [26,27] include indoor and outdoor air pollution, which again disproportionately affects those in poverty. High environmental temperatures are also associated with growth faltering, which is becoming a pervasive threat with the growing impacts of the climate crisis with similarly disproportionate impacts [28].

Maternal mental health also plays a substantial role [29–31] in growth faltering and is itself linked to poverty and related biological and psycho-emotional life stressors including household food insecurity. Genetic anomalies and developmental conditions, such as cleft palate [32], and heart conditions [33] are also drivers of growth faltering. Some of these conditions then also lead to maternal poor mental health, so this relationship can be circular [30]. Conversely, social support is a key mitigator of adverse mental health outcomes [30], demonstrating the importance of social dynamics in caring for faltering infants. Beyond the other social determinants of adverse birth outcomes mentioned above, additional social drivers of growth may also entail broader gender power dynamics, with women's empowerment and agency linked to infant and child growth [34,35].

Previous work has identified substantial gaps in the availability of interventions that effectively address growth faltering during infancy and early childhood. A key element that such interventions would need to consider is whether mothers have an enabling breastfeeding environment in which skin to skin contact, early initiation of and exclusive breastfeeding are supported [2,36]. Moreover, when mothers breastfeeding confidence is undermined and there is a lack of breastfeeding support, they are more likely to introduce supplements or end breastfeeding prematurely, increasing risk of infection [37,38], and hence growth faltering. Furthermore, even when supplementation with commercial milk formula (CMF) and/or specially formulated formulas and foods have the potential for increasing growth, there are also potential risks both from the greater velocity of catch-up growth and corresponding metabolic consequences as well as from greater risk of infection [39].

Specific evidence on elements of successful growth interventions is often lacking. In large cross-sectional survey in Africa, positive maternal education (years of schooling) and prenatal visits were associated with improved linear growth even in conditions of poverty, highlighting protective factors that could be the basis of future interventions [40]. Critically, kangaroo mother care (KMC) and best breastfeeding practices are rarely fully addressed in available studies [6,24,36,41]. A recent scientific consultation addressing growth faltering in India highlighted some of these research gaps as well as the inadequate follow-up after transitioning from facility-based services to the home-setting [42]. Further strategies that are now standard of care in clinical settings in many middle- and high-income countries to preserve human milk feeding in very preterm and medically compromised infants such as increasing feeding volumes, nutrient-enriching expressed human milk with commercial fortifiers or less costing nutrient modulars (e.g. protein, calcium, phosphorus) or use of donor milk have received little consideration in the context of the late-preterm or term-born infant with growth faltering [6,41]. Additionally, feeding interventions may not be adequate on their own to address growth faltering since some of the drivers require other medical treatment and/or addressing broader environmental factors.

This systematic review was commissioned by the World Health Organization to address questions about optimal supplementation of infants under 6 months of age experiencing growth faltering to inform discussion of the guidelines on treatment of infants and young children with wasting and/or oedema. The review did not examine nutritional interventions aimed at breastfeeding mothers of growth faltering infants. The framework for the review and related recommendations are to be understood in the context of best practices for infant feeding and care, which entail appropriate protection, promotion and support for breastfeeding.

## Methods

This study was a systematic review and meta-analysis following the US National Academies of Sciences, Engineering, and Medicine (NASEM) guidelines. The protocol was registered with PROSPERO (CRD42022350150). The review questions were:

1. In infants under 6 months with growth faltering, which criteria best determine if and when an infant should be given a supplemental milk (in addition to breastmilk if the infant is breastfed)?

2. In infants under 6 months with growth faltering meeting criteria in question 1, what is the most effective supplemental milk and for how long should it be given?

Supplemental milk was defined as donor human milk, fortified donor human milk, fortified expressed mothers' own milk, human milk from wet nurse, commercial infant formula, nutrient enriched commercial infant formula, F-75, F-100, or diluted F-100.

Searches were conducted to examine interventions addressing infants under 6 months with growth faltering that used supplemental milks as part of their treatment. Searches were framed as broadly as possible to capture any potentially relevant literature related to the initial questions. Primary outcomes of interest were anthropometry, mortality, clinical deterioration, morbidity, relapse, readmission, and non-response to treatment. Randomized controlled trials, cluster-randomized trials, and quasi-experimental studies/non-randomized controlled studies were included. Qualitative studies, reviews, and systematic reviews, as well as observational studies were excluded. All countries and settings were included.

## Searches

The following databases was searched electronically in August 2022: MEDLINE, Embase, CINAHL, Web of Science, Cochrane Library, Global Index Medicus, State of Acute Malnutrition, Emergency Nutrition Network (Field Exchange), eLENA (WHO), UNICEF, FAO, WFP, FAQR, and ClinicalTrials.gov, ISRCTN registry, ICMJE registry, World Health Organization (WHO) International Clinical Trials Registry Platform. The initial searches were designed, tested, and run by a well-trained team that includes public health scientists with expertise in breastfeeding research and a medical librarian. The search strategy developed during preliminary searches are presented in S1 Appendix. Search results were combined and de-duplicated in EndNote by the JHU Welch Library Team. Experts in the field were consulted to identify additional articles to be included.

## Data extraction

The study was carried out following the guidance of Preferred Reporting Items for Systematic Reviews and Meta-Analyses (PRISMA) [43]. Search results were exported into EndNote, de-duplicated, and uploaded into Covidence systematic review software for screening.

Two research assistants independently screened for potential inclusion of all titles and abstracts identified through the search. Any discrepancies were resolved by consensus or contacting a third author. Attempts were made to contact authors of included studies to obtain clarifications or additional data. Following title and abstract screening, full texts were independently screened for inclusion. Any disagreements were resolved through discussion or by consulting a third review author, if required. Reasons of exclusion were recorded for all the studies excluded at the stage of full text screening. The results of the search are presented using the PRISMA flow diagram.

Structured forms were used for data extraction of key variables including study characteristics and outcomes in a standardized data collection form. Two review authors independently extracted data and discrepancies were resolved through discussion until consensus had been achieved or by consulting a third reviewer, if required.

## Risk of bias (quality) assessment

The updated Cochrane risks of bias tool, ROB-2, was used for randomized trials [44]. Two independent authors assessed quality of all eligible studies and disagreements were resolved by consensus or contacting a third author. For all data extraction and study quality assessments two researchers were standardized against each other with screening, data extraction and to assess the quality of each study.

## Data synthesis

Statistical heterogeneity was assessed using $\tau^2$ (among study variance), $I^2$ (degree of inconsistency), and significance of the $\chi^2$ (chi-square) test; and visual inspection of forest plots. Based on prior clinical knowledge, we expected clinical and methodological heterogeneity in included studies and therefore, attempted to explain any observed statistical heterogeneity. The Grading of Recommendations, Assessment, Development and Evaluations (GRADE) [45] methodology was used to assess the certainty of the evidence by outcome. Two independent authors assessed quality of all eligible studies and disagreements were resolved by consensus or contacting a third author. Evidence profiles were generated via GRADE Pro Software.

## Results

A total of 10405 references were identified by the searches and imported for screening (Fig 1). Of these, 3015 duplicates were removed. 7390 studies were screened against title and abstract, 7163 studies were excluded, and 227 studies were assessed for full text eligibility. A total of 158 were excluded because the studies were not fully reported (n = 50 study protocols or conference abstracts), had the wrong patient population (n = 30), the wrong study design (n = 26), the wrong intervention (n = 20), the wrong outcomes (n = 14), or were duplicates

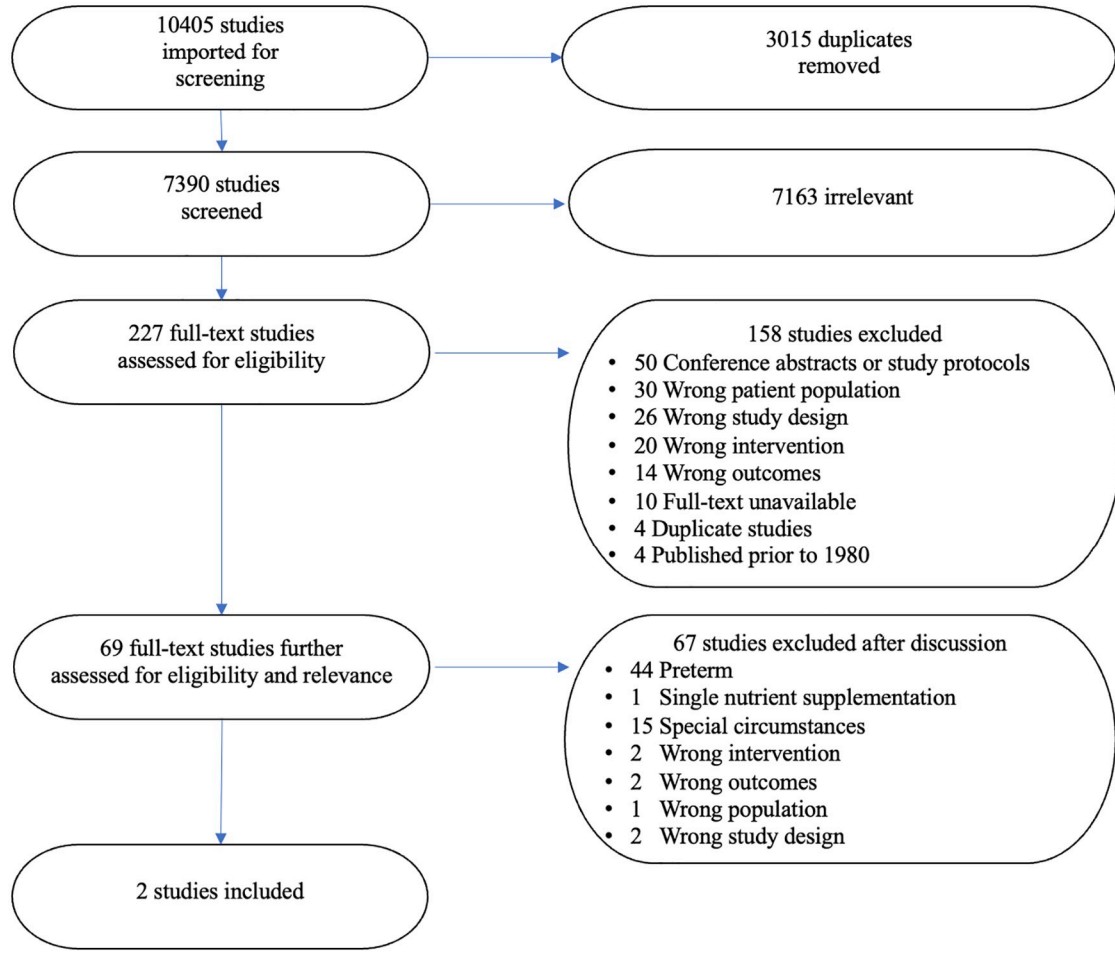

**Fig 1. PRISMA flow chart.**

(n = 4). Full texts were unavailable for 10 studies. Additionally, 4 were carried out prior to 1980 after which significant changes took place in the composition of infant formulas, yielding 69 studies that were published between 1980 and August 2022, when the searches were carried out.

Of the remaining 69 studies, 60 were excluded because they focused on: 1) preterm, predominately hospitalized infants primarily from high income countries 2) infants who received single nutrient supplementation, 3) infants with other special circumstances (e.g., recovering from cardiac surgery) (S2 Appendix). Two studies within the infants with special circumstances category were examined further for relevant contextual information (Clarke et al [46] and Evans et al [47]) but were ultimately excluded because the study sample was principally comprised of infants with a significant medical condition known to elevate nutritional requirements and impact growth. Only one infant (of 30) received breastmilk in a previous feed in Evans et al, while Clarke et al did not report any breastfeeding or receipt of breast milk, making it likely participant infants were nearly all exclusively formula-fed in both studies during the treatment period.

Evans et al [47] enrolled 30 growth faltering infants with a median age of 14.5 weeks (range 2–43 weeks) who were receiving care at a large British specialty referral hospital, 73% (n = 22) of whom had cardiac defects. They compared a ready to feed high-energy (1kcal/ml) nutrient-dense (concentrated macro- and micronutrients) infant formula at full strength (25 mOsm kg) 1 H2O) from day one with stepwise introduction of the same ready to feed infant formula but in a diluted form and graded to full strength over three days; infants were studied for a duration of 2 weeks. While younger infants in the high energy nutrient dense-formula group had an initial increased number of stools, no differences in feeding tolerance nor growth were observed between treatment arms.

Clarke et al [46] enrolled 60 growth faltering infants with a median age of 5 weeks (range 2–31) with cardiac lesions, cystic fibrosis, or other organic causes from Birmingham Children's Hospital in the UK. This open RCT compared two specialty infant formulas, 'high-energy nutrient-dense formula' and a 'high energy only-supplemented formula,' for a period of 6 weeks. The energy concentration of both formulas was 1 kcal/ml; energy was supplied in the "high energy only-supplemented formula" by adding glucose whereas in the "high energy nutrient-dense formula" it was supplied by protein and glucose. The protein concentration of the high energy only and high energy nutrient dense formulas was 1.4 and 2.6 grams/100 kcal, respectively. Infants in the "high energy nutrient-dense formula" group had higher intakes of protein, vitamins and minerals and blood urea nitrogen levels suggesting more favorable protein balance; further there was less fall off of length for age-z-scores compared to the "high energy only-supplemented" group.

An additional seven studies were determined to have the wrong intervention, outcomes, population, or study design (S2 Appendix). While initial searches undertaken were purposefully broad, in the full text review of manuscripts it became clear that most included studies focused on preterm babies in high-income settings and/or addressed special populations with particular medical conditions. These studies did not address treatment and recovery from malnutrition, which was the purpose of this review. Therefore, after careful consideration 67 additional studies were excluded, yielding two final studies for consideration.

Two studies ultimately generated direct evidence about the review questions (Islam et al [48] and Wilkinson [49]). Islam et al [48] was published in the peer reviewed literature, while Wilkinson [49] was initially identified in the grey literature, and the full report was provided by the author to the review team.

## Included studies

**Islam et al (2020) [48].** Islam et al [48] was a three-armed randomized controlled trial carried out in Bangladesh in 2012–2015 comparing F-100 (1 kcal/ml, 2.9 g protein/ 100ml; n = 50), diluted F-100 (0.77 kcal/ml, 2.2 g protein/100 ml; DF-100, n = 52) and standard infant formula (0.70 kcal/ml, 1.6 g protein/100 ml; IF, n = 50) in the context of infant rehabilitation from severe acute malnutrition (SAM) (Table 1). Islam et al (2020) [48] included infants under 6 months of age admitted for diarrheal illness with weight-for-length z-score of <-3 and/or bipedal edema that were stabilized in a ward for infants requiring long term specialized care less than nine days (free of edema, with cessation of vomiting, fewer than three watery stools /day and no clinical signs of infection). Infants in the three study arms were comparable in most biomedical, socio-economic and demographic characteristics, however, those in the F-100 group were significantly more wasted at the time of enrollment in the trial compared to infants in the other two study arms (p = 0.02).

Once stabilized per the above criteria, infants were provided with F-100, DF-100 or standard IF. During the first two days infants received the study diet of 130 ml/kg/day in 12 feeds and then increased by 10 ml per feed until some remained. They were fed every three hours thereafter. Infants were discharged when they gained 15% of their weight or had edema-free weight-for-length z-score $\geq -2$. Mothers who were breastfeeding their participant infants were encouraged to continue doing so. Notably, only approximately half of the participants were breastfed on entry into the trial. Additionally, those who were breastfed received a very small proportion of their feeds from breastfeeding (e.g., less than 10% of total kcal/kg/day, on average). Water was also offered between feeds.

If infants became ill during the treatment, they were returned to the ward for infants requiring long term specialized care or another ward for short-term specialized care, fed F-75 (0.75 kcal/ml and 0.9 g protein/100 ml), and returned to the trial once they recovered from illness. A third of participants (n = 52; F-100 n = 19; DF-100 n = 20; IF n = 13) experienced illness (diarrhea and pneumonia) that necessitated transfer to one of the longer term or specialty wards.

**Table 1. Summary of included studies (Islam et al 2020 [48], Wilkinson 2008 [49]).**

| First author, year. Location/setting | Design | Sample description | Interventions | Outcomes | Risk of bias |
|---|---|---|---|---|---|
| Islam et al 2020 [48] Dhaka, Bd Nutrition Rehabilitation Unit 2012–2015 | RCT, double blinded | n = 153 infants with diarrheal illness. <6mos, weight-for-length z-score <-3 and/or bipedal oedema. After stabilization: free of oedema with no clinical signs of infection. | After stabilization phase with F-75, randomized to 130 ml/kg/day in 12 (2-hourly) feeds, and then 10 ml more at each feed for the first 2 days, and then fed at 3-hourly intervals: **lntx. 1**: F-100 (n = 50) **Intx. 2**: Diluted F-100 (n = 52) **lntx. 3**: Infant formula (n = 51) Mothers who breastfed their hospitalized infants were encouraged to continue. If infant became ill -> transferred to specialty care ward and fed F-75 instead of assigned diet. After recovery, resumed allocated diet. | Main: Rate of weight gain (g/kg/day) in rehabilitation phase. Secondary: potential renal solute load (PRSL) (mOsmol/l) renal solute load (RSL), (mOsmol/l), serum sodium (mmol/l), urinary specific gravity; study diet intake (g/day) and (kcal/day), duration of stay at the NRU (in days) | High |
| Wilkinson, 2008 [49] South Kivu, DRC 2007–2008 | RCT | n = 146 infants. <6 mos at admission, currently breastfed, with failure of effective breastfeeding, or infant too weak to suckle effectively, or infant is not gaining weight at home. No oedema, or other conditions that interfered with ability to breastfeed. | Supplementary quantities of artificial milk were given every 3 hrs: Control: Diluted F-100 Intervention: Generic infant formula for children 0–12 months. Frequent breastfeeding and/or expressed breastmilk was encouraged for all infants. | Primary: rate of weight gain (g/kg/day) and mortality. Secondary: weight gain (g), duration of treatment (days), recovery, and defaulting. | High |

The number of infants transferred did not statistically differ across the three arms (p = 0.29). No information was provided about the length of treatment in the two other wards or additional details of comparability upon returning to the trial.

**Wilkinson (2008) [49].** Wilkinson's (2008) [49] study was a two-armed study comparing DF-100 (0.73 kcal/ml, 1.8 g protein/100 ml, n = 72) and a standard infant formula (0.77 kcal/ml, 1.95 g/100 ml; IF, n = 74) in the context of a Management of Acute Malnutrition in Infants (MAMI) project (Table 1). This study was designed to improve care for acutely malnourished infants under 6 months in Niger and two sites in the Democratic Republic of Congo (DRC) in 2007–2008. It is important to note that data from Niger were ultimately not presented in the final analysis due to small sample size and lack of adherence to the protocol. Due to fewer than expected visits to Therapeutic Feeding Centers (TFCs) during the study period, the study was not adequately powered to detect differences in treatment.

Wilkinson (2008) [49] included infants under six months free of any oedema, with maternal report of breastfeeding failure or not gaining weight at home. No blinding was possible due to the appearance and preparation of the supplemental milks. All mothers of participants were breastfeeding or were engaging in relactation and received lactation support; though this was not quantitated in the report. They used the supplementary suckling technique (SST) to stimulate relactation while receiving therapeutic feeding whenever possible. Supplementary feeds were reduced by half when infants were gaining 20 g/day. Finally, supplementation was stopped when infants maintained a weight gain of 10g/day for 3 days in a row. Infants were discharged when they were able to maintain weight gain for 3–5 days. If infants became ill, they were transferred to a hospital and continued to receive supplemental milk. Breastfeeding observations were not continued during hospitalization due to lack of staff. The author's description indicates that the ill infants who were transferred to the hospital remained in the trial although many of whom later died; it is not clear, however, how well protocols were followed for this group. The author notes that pediatric and neonatal services were limited and weak at the hospitals to which ill infants were transferred, with no specialized neonatal care for very young infants.

**Attrition, final sample size, and risk of bias.** Islam et al (2020) [48] experienced high attrition of 24 infants (16%) leaving the trial before discharge criteria were met across all three arms. Attrition did not statistically differ across the three study arms. Supplementary data suggest that 8 additional infants dropped out of the study (see Islam et al [48] Supplementary Table 2). One infant in the F-100 died due to a hospital acquired infection. Consequently, only 36/52 (F-100), 42/52 (DF-100), 42/50 (IF) infants from the three groups, respectively, completed the study to discharge. Islam et al [48] was rated at high risk of bias according to the ROB-2 assessment due to non-adherence to the assigned intervention and loss to follow-up (see Tables 1–4 for details).

The Wilkinson (2008) [49] trial also experienced high rates of infants leaving the trial prior to meeting discharge criteria (DF-100 n = 13; IF n = 10) (no difference between study arms). Additionally, there was a large number of deaths in both arms (DF-100 n = 12; IF n = 11) (no difference between study arms). One infant from each group was transferred out of the study. A total of 48 infants were recovered in the DF-100 and 50 in the IF group, respectively. These findings did not differ by intervention. Wilkinson's (2008) [49] study was rated at high risk of bias according to the ROB-2 assessment due to unblinding and loss to follow-up (see Tables 1–4 for details).

## Outcomes of interest

Due to the small number of studies identified by the search, the initial questions for the review could not be fully assessed. We present the outcomes of interest that are available in three

**Table 2. GRADE Evidence Profile: Diluted F-100 compared to infant formula for infants <6 months of age with severe wasting and/or oedema Author(s): Tomori et al.** Question: Diluted F-100 compared to infant formula for infants <6 months with severe wasting and/or oedema.

| No of studies | Study design | Risk of bias | Inconsistency | Indirectness | Imprecision | Other considerations | diluted F-100 | infant formula | Relative (95% CI) | Absolute (95% CI) | Certainty | Importance |
|---|---|---|---|---|---|---|---|---|---|---|---|---|
| | | | **Certainty assessment** | | | | **No of patients** | | **Effect** | | **Certainty** | **Importance** |
| Rate of weight gain (g/kg/day), during the rehabilitation phase (Islam et al 2020) [48] or from admission until discharge in infants who recovered (Wilkinson 2008) [49] | | | | | | | | | | | | |
| 2[a,b] | randomized trials | serious[c] | not serious | serious[d] | serious[e] | none | 100 | 101 | - | MD **1.72 higher** (1.39 lower to 4.83 higher) | Very low | CRITICAL |
| Body weight (kg), at the end of the rehabilitation phrase, total duration of stay unclear (Islam et al 2020) [48] | | | | | | | | | | | | |
| 1 | randomized trials | serious[f] | not serious | serious[d] | serious[g] | none | 52 | 51 | - | MD **0.08 higher** (0.29 lower to 0.45 higher) | Very low | CRITICAL |
| Length (cm), at the end of the rehabilitation phrase, total duration of stay unclear (Islam et al 2020) [48] | | | | | | | | | | | | |
| 1 | randomized trials | serious[f] | not serious | serious[d] | not serious | none | 52 | 51 | - | MD **0.3 lower** (2.03 lower to 1.43 higher) | Low | CRITICAL |
| WLZ, at the end of the rehabilitation phase, total duration of stay unclear (Islam et al 2020) [48] | | | | | | | | | | | | |
| 1 | randomised trials | serious[f] | not serious | serious[d] | serious[h] | none | 48 | 49 | - | MD **0.3 higher** (0.08 lower to 0.68 higher) | Very low | CRITICAL |
| WAZ, at the end of the rehabilitation phrase, total duration of stay unclear (Islam et al 2020) [48] | | | | | | | | | | | | |
| 1 | randomised trials | serious[f] | not serious | serious[d] | serious[h] | none | 50 | 49 | - | MD **0.1 lower** (0.4 lower to 0.6 higher) | Very low | CRITICAL |
| LAZ, at the end of the rehabilitation phrase, total duration of stay unclear (Islam et al 2020) [48] | | | | | | | | | | | | |
| 1 | randomized trials | serious[f] | not serious | serious[d] | serious[g] | none | 52 | 51 | - | MD **0** (0.65 lower to 0.65 higher) | Very low | CRITICAL |
| Mortality (Islam et al 2020 [48], Wilkinson 2008 [49]) | | | | | | | | | | | | |
| 2 | randomised trials | serious[c] | not serious | serious[d] | serious[i] | none | 12/126 (9.5%) | 11/123 (8.9%) | **RR 1.06** (0.49 to 2.32) | **5 more per 1,000** (from 46 fewer to 118 more) | Very low | IMPORTANT |
| Clinical deterioration defined by development of any danger signs–not measured | | | | | | | | | | | | |
| | | | | | | | | | | | | IMPORTANT |
| Morbidity or recovery from co-morbidity: "hospital-acquired infection" (Islam et al 2020) [48] | | | | | | | | | | | | |
| 1[j] | randomized trials | serious[f] | not serious | serious[d] | serious[k] | none | 20/52 (38.5%) | 13/51 (25.5%) | **RR 1.51** (0.84 to 2.70) | **130 more per 1,000** (from 41 fewer to 433 more) | Very low | IMPORTANT |
| Relapse–not measured | | | | | | | | | | | | |
| | | | | | | | | | | | | IMPORTANT |
| Readmission–not measured | | | | | | | | | | | | |
| | | | | | | | | | | | | IMPORTANT |
| Non-response–not measured | | | | | | | | | | | | |
| | | | | | | | | | | | | IMPORTANT |
| Potential renal solute load (mOsmol/day), at discharge (Islam et al 2020) [48] | | | | | | | | | | | | |
| 1 | randomised trials | serious[f] | not serious | serious[d] | serious[l] | none | 52 | 51 | - | MD **21.5 higher** (6.4 higher to 36.5 higher) | Very low | |
| Renal solute load (mOsmol/day, at discharge (Islam et al 2020) [48] | | | | | | | | | | | | |
| 1 | randomised trials | serious[f] | not serious | serious[d] | serious[l] | none | 52 | 51 | - | MD **16.2 higher** (1.5 higher to 30.8 higher) | Very low | |
| Serum sodium (mmol/L), day 7 (Islam et al 2020) [48] | | | | | | | | | | | | |
| 1 | randomised trials | serious[f] | not serious | serious[d] | not serious | none | 44 | 41 | - | MD **0.3 lower** (1.3 lower to 0.7 higher) | Low | |

(*Continued*)

**Table 2.** (*Continued*)

| No of studies | Study design | Risk of bias | Inconsistency | Indirectness | Imprecision | Other considerations | diluted F-100 | infant formula | Relative (95% CI) | Absolute (95% CI) | Certainty | Importance |
|---|---|---|---|---|---|---|---|---|---|---|---|---|
| | | | | **Certainty assessment** | | | | **No of patients** | **Effect** | | **Certainty** | **Importance** |
| Serum chloride (mmol/L). day 7 (Islam et al 2020) [48] | | | | | | | | | | | | |
| 1 | randomised trials | serious[f] | not serious | serious[d] | not serious | none | 44 | 41 | - | MD **0.2 higher** (1.21 lower to 1.61 higher) | Low | |

CI: Confidence interval, MD: Mean difference, RR: Risk ratio

Explanations

[a]. One study (Islam et al 2020) [48] included infants under six months admitted for diarrheal illness with weight-for-length z-score of <3 and/or bilateral oedema that were stabilized in less than 9 days (free of oedema, with cessation of vomiting, fewer than 3 watery stools/day and no clinical signs of infection). However, if any of the infants became ill, they were transferred back to a Longer Stay or Special Care Ward and fed F-75; after recovery they were returned to the study diet. The other study (Wilkinson 2008) [49] included infants under six months who were too weak or feeble to suckle effectively, with breastfeeding failure, not gaining weight at home, or with weight-for-length below 70% of the NCHS medial. Infants with oedema were excluded.

[b]. One study (Islam et al 2020) [48] fed infants every 2 hrs with F-75 during the stabilization phase. For the first two days of the rehabilitation phase they received 130 ml/kg/day of F-100 or infant formula in 12 (2-hr) feeds and then 10 ml more at each feed until some was left (ad libitum). Water was freely available. The other study (Wilkinson 2008) [49] gave supplementary quantities of artificial milk every 3 hours, using the SST technique; frequent breastfeeding was encouraged. When infants gained > = 20g/day for three consecutive days, the milk supplement was reduced to 1/2; when infants maintained weight gain > = 10g/day on reduced quantities for three consecutive days, the milk supplement was stopped altogether and infants received only breast milk; when infants maintained weight gain of 10g/day for three to five consecutive days, infants were considered recovered and were discharged.

[c]. Serious risk of bias. One of the two studies (Wilkinson 2008) [49] had a high risk of bias (due to unblinding, loss to follow-up) while the other study (Islam et al 2020) [48] had a high risk of bias (due to non-adherence to the assigned intervention, loss to follow-up).

[d]. Serious indirectness: The supplemental milks in the Islam et al 2020 [48] trial were given only to infants with severe wasting and/or oedema who had already been stabilized and given F-75 prior to the rehabilitation phase, and if they became ill they were given F-75 again before resuming one of the three study milks. See footnotes a and b for additional details of the study populations and intervention approaches.

[e]. Serious imprecision. 95% CI around the absolute effect crosses the null, and the effect ranges from trivial harm to moderate benefit.

[f]. Serious risk of bias: The only study (Islam et al 2020) [48] had a high risk of bias (due to non-adherence to the assigned intervention, loss to follow-up).

[g]. Serious imprecision: 95% CI around the absolute effect crosses the null, and the effect ranges from small harm to small benefit.

[h]. Serious imprecision: 95% CI around the absolute effect crosses the null, and the effect ranges from trivial harm to small benefit.

[i]. Serious imprecision: 95% CI around the absolute effect crosses the null, and the effect ranges from small benefit to moderate harm.

[j]. "Fifty-two infants temporarily went to the Longer Stay Ward or Special Care Ward for management of hospital-acquired infection during the intervention period. The numbers were 19, 20 and 13, respectively, in the F-100, diluted F-100, and infant formula groups (P = 0.29)"

[k]. Serious imprecision: 95% CI around the absolute effect crosses the null, and the effect ranges from small benefit to serious harm.

[l]. Serious imprecision: 95% CI around the absolute effect does not cross the null, however the 95% CI includes both small to moderate harm.

GRADE tables (Tables 2–4 below). Table 2 combines the comparison shared by the two included studies (DF-100 compared with IF). Detailed information about the populations and interventions is presented in Table 2 footnotes a) and b) to help contextualize the findings.

Additionally, Table 3 presents the F-100 v. IF comparison, and Table 4 presents the F-100 v DF-100 comparison, both from Islam et al [48].

Rate of weight gain was the only outcome assessed for DF-100 v IF in both studies. Islam et al (2020) [48] found significantly higher rates of weight gain for both F-100 (mean difference 4.6; 95% CI 1.5–7.6; p = 0.004) and DF-100 (mean difference 3.1; 95% CI 0.6–5.5; p = 0.015) when compared with IF, respectively. In contrast, Wilkinson (2008) [49] did not find significant differences in rate of weight gain between DF-100 and IF (0.1 g/kg/day greater for IF-100, 95% CI: -3.3, 3.5).

The weight gain outcome (g/kg/day) was meta-analyzed with the two studies (Islam et al 2020 [48] and Wilkinson 2008 [49]) using a random-effects model (Fig 2). The meta-analysis

**Table 3. GRADE Evidence Profile: F-100 compared to infant formula for infants <6 months of age with severe wasting and/or oedema Author(s): Tomori et al.**
Question: F-100 compared to infant formula for infants <6 months with severe wasting and/or oedema.

| No of studies | Study design | Risk of bias | Inconsistency | Indirectness | Imprecision | Other considerations | F-100 | infant formula | Relative (95% CI) | Absolute (95% CI) | Certainty | Importance |
|---|---|---|---|---|---|---|---|---|---|---|---|---|
| | | | Certainty assessment | | | | No of patients | | Effect | | | |
| Rate of weight gain (g/kg/day), during the rehabilitation phase (Islam et al 2020) [48] | | | | | | | | | | | | |
| 1 | randomized trials | serious[a] | not serious | serious[b] | serious[c] | none | 50 | 51 | - | MD **4.6 higher** (1.5 higher to 7.6 higher) | Very low | CRITICAL |
| Body weight (kg), at the end of the rehabilitation phrase, total duration of stay unclear (Islam et al 2020) [48] | | | | | | | | | | | | |
| 1 | randomized trials | serious[a] | not serious | serious[b] | serious[d] | none | 50 | 51 | - | MD **0.08 lower** (0.43 lower to 0.27 higher) | Very low | CRITICAL |
| Length (cm), at the end of the rehabilitation phrase, total duration of stay unclear (Islam et al 2020) [48] | | | | | | | | | | | | |
| 1 | randomized trials | serious[a] | not serious | serious[b] | not serious | none | 50 | 51 | - | MD **0.4 lower** (1.95 lower to 1.15 higher) | Low | CRITICAL |
| WLZ, at the end of the rehabilitation phase, total duration of stay unclear (Islam et al 2020) [48] | | | | | | | | | | | | |
| 1 | randomised trials | serious[a] | not serious | serious[b] | serious[e] | none | 49 | 49 | - | MD **0.1 lower** (0.58 lower to 0.38 higher) | Very low | CRITICAL |
| WAZ, at the end of the rehabilitation phrase, total duration of stay unclear (Islam et al 2020) [48] | | | | | | | | | | | | |
| 1 | randomised trials | serious[a] | not serious | serious[b] | serious[e] | none | 49 | 49 | - | MD **0.1 lower** (0.6 lower to 0.4 higher) | Very low | CRITICAL |
| LAZ, at the end of the rehabilitation phrase, total duration of stay unclear (Islam et al 2020) [48] | | | | | | | | | | | | |
| 1 | randomized trials | serious[a] | not serious | serious[b] | serious[f] | none | 52 | 51 | - | MD **0** (0.61 lower to 0.61 higher) | Very low | CRITICAL |
| Mortality (Islam et al 2020) [48] | | | | | | | | | | | | |
| 1 | randomised trials | serious[a] | not serious | serious[b] | serious[f] | none | 1/50 (2.0%) | 0.1/51 (0.2%) | RR **3.06** (0.13 to 73.35) | **4 more per 1,000** (from 2 fewer to 142 more) | Very low | IMPORTANT |
| Clinical deterioration defined by development of any danger signs–not measured | | | | | | | | | | | | |
| | | | | | | | | | | | | IMPORTANT |
| Morbidity or recovery from co-morbidity: "hospital-acquired infection" (Islam et al 2020) [48] | | | | | | | | | | | | |
| 1[g] | randomised trials | serious[a] | not serious | serious[b] | serious[h] | none | 19/50 (38.0%) | 13/51 (25.5%) | RR **1.49** (0.83 to 2.68) | **125 more per 1,000** (from 43 fewer to 428 more) | Very low | IMPORTANT |
| Relapse–not measured | | | | | | | | | | | | |
| | | | | | | | | | | | | IMPORTANT |
| Readmission–not measured | | | | | | | | | | | | |
| | | | | | | | | | | | | IMPORTANT |
| Non-response–not measured | | | | | | | | | | | | |
| | | | | | | | | | | | | IMPORTANT |
| Potential renal solute load (mOsmol/day), at discharge (Islam et al 2020) [48] | | | | | | | | | | | | |
| 1 | randomised trials | serious[a] | not serious | serious[b] | serious[i] | none | 50 | 51 | - | MD **44.6 higher** (27.2 higher to 62.1 higher) | Very low | |
| Renal solute load (mOsmol/day, at discharge (Islam et al 2020) [48] | | | | | | | | | | | | |
| 1 | randomised trials | serious[a] | not serious | serious[b] | serious[i] | none | 50 | 51 | - | MD **40.2 higher** (23.5 higher to 57 higher) | Very low | |
| Serum sodium (mmol/L), day 7 (Islam et al 2020) [48] | | | | | | | | | | | | |

*(Continued)*

**Table 3.** (Continued)

| No of studies | Certainty assessment | | | | | | No of patients | | Effect | | Certainty | Importance |
|---|---|---|---|---|---|---|---|---|---|---|---|---|
| | Study design | Risk of bias | Inconsistency | Indirectness | Imprecision | Other considerations | F-100 | infant formula | Relative (95% CI) | Absolute (95% CI) | | |
| 1 | randomised trials | serious[a] | not serious | serious[b] | not serious | none | 36 | 41 | - | MD **0.6 lower** (1.7 lower to 0.5 higher) | Low | |
| Serum chloride (mmol/L). day 7 (Islam et al 2020) [48] | | | | | | | | | | | | |
| 1 | randomised trials | serious[a] | not serious | serious[b] | not serious | none | 36 | 41 | - | MD **1.4 higher** (0.26 lower to 3.06 higher) | Low | |

CI: Confidence interval, MD: Mean difference, RR: Risk ratio

Explanations

[a]. Serious risk of bias. The only study (Islam et al 2020) [48] had a high risk of bias (due to non-adherence to the assigned intervention, loss to follow-up).

[b]. Serious indirectness: The supplemental milks in the Islam et al 2020 [48] trial were given only to infants with severe wasting and/or oedema who had already been stabilized and given F-75 prior to the rehabilitation phase, and if they became ill they were given F-75 again before resuming one of the three study milks. See Table 2 footnotes a and b for additional details of the study populations and intervention approaches.

[c]. Serious imprecision. 95% CI around the absolute effect does not cross the null, however the 95% CI includes both small to moderate benefit.

[d]. Serious imprecision: 95% CI around the absolute effect crosses the null, and the effect ranges from trivial harm to trivial benefit.

[e]. Serious imprecision: 95% CI around the absolute effect crosses the null, and the effect ranges from small harm to trivial benefit.

[f]. Serious imprecision: 95% CI around the absolute effect crosses the null, and the effect ranges from small benefit to serious harm.

[g]. "Fifty-two infants temporarily went to the Longer Stay Ward or Special Care Ward for management of hospital-acquired infection during the intervention period. The numbers were 19, 20 and 13, respectively, in the F-100, diluted F-100, and infant formula groups (P = 0.29)."

[h]. Serious imprecision: 95% CI around the absolute effect crosses the null, and the effect ranges from small benefit to serious harm.

[i]. Serious imprecision: 95% CI around the absolute effect does not cross the null, however the 95% CI includes both small to moderate harm.

estimate of the mean difference in weight gain between IF and DF-100 was 1.72 g/kg/day, which was not statistically significant (p = 0.278); 95% CI: -1.39, 4.83. There was some evidence of moderate heterogeneity across studies, although it was not statistically significant; Q(1) = 2.30, p = 0.130; $I^2$ = 56.5.

Total weight gain at discharge was only reported for Wilkinson (2008) [49] and did not significantly differ between the two groups (non-significantly higher weight gain for DF-100 compared with IF; 63.8 g (95% CI: -53.4, 181).

There was one death in Islam et al's (2020) [48] study due to a hospital acquired infection, but it was deemed unrelated to the infant's diet. Mortality was high in both arms of the Wilkinson [49] study (DF-100 n = 12; IF n = 11) but did not differ by treatment arm. Infants who died were all under 60 days old at admission; 43% (10/23) were less than 30 days old at admission, and 56% (13/23) were less than 60 days old at admission.

Islam et al (2020) [48] reported faster recovery for the F-100 group than for the IF group (p = 0.043), but the rate of recovery for the F-100 group was not faster than for infants who received DF-100 (p = 0.09). There were no differences in rate of recovery or duration of treatment from admission to discharge between the two arms of the Wilkinson (2008) [49] study overall. However, there were significant differences between the two centers in duration of treatment, with significantly shorter treatment from admission to discharge for IF compared with DF-100 in Baraka (4.7 days 95% CI: -10.7, 1.3) and significantly longer treatment from admission to discharge for the IF arm compared with DF-100 in Uvira (3.5 days 95% CI: -1.5, 8.5).

Islam et al (2020) [48] also examined additional outcomes to assess the safety of F-100 based on previous concerns that it would result in high renal solute load (RSL) and risk of

**Table 4. GRADE Evidence Profile: F-100 compared to diluted F-100 for infants <6 months of age with severe wasting and/or oedema Author(s): Tomori et al.**
Question: F-100 compared to diluted F-100 for infants <6 months with severe wasting and/or oedema.

| No of studies | Study design | Risk of bias | Inconsistency | Indirectness | Imprecision | Other considerations | F-100 | diluted F-100 | Relative (95% CI) | Absolute (95% CI) | Certainty | Importance |
|---|---|---|---|---|---|---|---|---|---|---|---|---|
| | | | | Certainty assessment | | | | No of patients | | Effect | Certainty | Importance |
| Rate of weight gain (g/kg/day), during the rehabilitation phase (Islam et al 2020) [48] | | | | | | | | | | | | |
| 1 | randomized trials | serious[a] | not serious | serious[b] | serious[c] | none | 50 | 52 | - | MD **1.5 higher** (1.7 lower to 4.8 higher) | Very low | CRITICAL |
| Body weight (kg), at the end of the rehabilitation phrase, total duration of stay unclear (Islam et al 2020) [48] | | | | | | | | | | | | |
| 1 | randomized trials | serious[a] | not serious | serious[b] | serious[d] | none | 50 | 52 | - | MD **0.16 lower** (0.57 lower to 0.25 higher) | Very low | CRITICAL |
| Length (cm), at the end of the rehabilitation phrase, total duration of stay unclear (Islam et al 2020) [48] | | | | | | | | | | | | |
| 1 | randomized trials | serious[a] | not serious | serious[b] | not serious | none | 50 | 52 | - | MD **0.1 lower** (1.96 lower to 1.76 higher) | Low | CRITICAL |
| WLZ, at the end of the rehabilitation phase, total duration of stay unclear (Islam et al 2020) [48] | | | | | | | | | | | | |
| 1 | randomised trials | serious[a] | not serious | serious[b] | serious[e] | none | 49 | 48 | - | MD **0.4 lower** (0.83 lower to 0.03 higher) | Very low | CRITICAL |
| WAZ, at the end of the rehabilitation phrase, total duration of stay unclear (Islam et al 2020) [48] | | | | | | | | | | | | |
| 1 | randomised trials | serious[a] | not serious | serious[b] | serious[e] | none | 49 | 50 | - | MD **0.2 lower** (0.75 lower to 0.35 higher) | Very low | CRITICAL |
| LAZ, at the end of the rehabilitation phrase, total duration of stay unclear (Islam et al 2020) [48] | | | | | | | | | | | | |
| 1 | randomized trials | serious[a] | not serious | serious[b] | serious[f] | none | 50 | 52 | - | MD **0** (0.74 lower to 0.74 higher) | Very low | CRITICAL |
| Mortality (Islam et al 2020) [48] | | | | | | | | | | | | |
| 1 | randomised trials | serious[a] | not serious | serious[b] | serious[f] | none | 1/50 (2.0%) | 0.1/52 (0.2%) | **RR 3.12** (0.13 to 74.78) | **4 more per 1,000** (from 2 fewer to 142 more) | Very low | IMPORTANT |
| Clinical deterioration defined by development of any danger signs–not measured | | | | | | | | | | | | |
| | | | | | | | | | | | | IMPORTANT |
| Morbidity or recovery from co-morbidity: "hospital-acquired infection" (Islam et al 2020) [48] | | | | | | | | | | | | |
| 1[g] | randomised trials | serious[a] | not serious | serious[b] | serious[h] | none | 19/50 (38.0%) | 20/52 (38.5%) | **RR 0.99** (0.60 to 1.62) | **4 fewer per 1,000** (from 154 fewer to 238 more) | Very low | IMPORTANT |
| Relapse–not measured | | | | | | | | | | | | |
| | | | | | | | | | | | | IMPORTANT |
| Readmission–not measured | | | | | | | | | | | | |
| | | | | | | | | | | | | IMPORTANT |
| Non-response–not measured | | | | | | | | | | | | |
| | | | | | | | | | | | | IMPORTANT |
| Potential renal solute load (mOsmol/day), at discharge (Islam et al 2020) [48] | | | | | | | | | | | | |
| 1 | randomised trials | serious[a] | not serious | serious[b] | serious[i] | none | 50 | 52 | - | MD **23.1 higher** (4.1 higher to 42.2 higher) | Very low | |
| Renal solute load (mOsmol/day, at discharge (Islam et al 2020) [48] | | | | | | | | | | | | |
| 1 | randomised trials | serious[a] | not serious | serious[b] | serious[i] | none | 50 | 52 | - | MD **24 higher** (5.2 higher to 42.9 higher) | Very low | |
| Serum sodium (mmol/L), day 7 (Islam et al 2020) [48] | | | | | | | | | | | | |

(*Continued*)

**Table 4.** (Continued)

| No of studies | Study design | Risk of bias | Inconsistency | Indirectness | Imprecision | Other considerations | F-100 | diluted F-100 | Relative (95% CI) | Absolute (95% CI) | Certainty | Importance |
|---|---|---|---|---|---|---|---|---|---|---|---|---|
| Certainty assessment | | | | | | | No of patients | | Effect | | | |
| 1 | randomised trials | serious[a] | not serious | serious[b] | not serious | none | 36 | 44 | - | MD **0.2 lower** (1.07 lower to 0.67 higher) | Low | |
| Serum chloride (mmol/L). day 7 (Islam et al 2020) [48] | | | | | | | | | | | | |
| 1 | randomised trials | serious[a] | not serious | serious[b] | not serious | none | 36 | 44 | - | MD **1.2 higher** (0.27 lower to 2.67 higher) | Low | |

CI: Confidence interval, MD: Mean difference, RR: Risk ratio

Explanations

[a]. Serious risk of bias. The only study (Islam et al 2020) [48] had a high risk of bias (due to non-adherence to the assigned intervention, loss to follow-up).

[b]. Serious indirectness: The supplemental milks in the Islam et al 2020[48] trial were given only to infants with severe wasting and/or oedema who had already been stabilized and given F-75 prior to the rehabilitation phase, and if they became ill they were given F-75 again before resuming one of the three study milks. See Table 2 footnotes a and b for additional details of the study populations and intervention approaches.

[c]. Serious imprecision. 95% CI around the absolute effect crosses the null, and the effect ranges from trivial harm to small benefit.

[d]. Serious imprecision: 95% CI around the absolute effect crosses the null, and the effect ranges from trivial harm to trivial benefit.

[e]. Serious imprecision: 95% CI around the absolute effect crosses the null, and the effect ranges from small harm to trivial benefit.

[f]. Serious imprecision: 95% CI around the absolute effect crosses the null, and the effect ranges from small benefit to serious harm.

[g]. "Fifty-two infants temporarily went to the Longer Stay Ward or Special Care Ward for management of hospital-acquired infection during the intervention period. The numbers were 19, 20 and 13, respectively, in the F-100, diluted F-100, and infant formula groups (P = 0.29)."

[h]. Serious imprecision: 95% CI around the absolute effect crosses the null, and the effect ranges from moderate harm to moderate benefit.

[i]. Serious imprecision: 95% CI around the absolute effect does not cross the null, however the 95% CI includes both small to moderate harm.

[j]. Serious imprecision: 95% CI around the absolute effect does not cross the null, however the 95% CI includes both small to moderate harm.

hyponatraemic dehydration. The study did not find evidence of these potential safety issues. The RSL was highest in the F-100 group (126.4 51.9 ± 51.9 mOsm/day) and was significantly different between F-100 and DF-100 as well as between F-100 and IF groups. However, sodium and urinary specific gravity were within normal range (Tables 2–4).

## Discussion

The Islam et al (2020) [48] trial recommended the use of undiluted F-100 as a simpler approach and did not find any safety concerns in the controlled, hospital environment of the trial. Although it should be noted that none of the infants in the trial had diarrhea, nor had dehydration. This is because in that trial infants who became ill with diarrhea, were temporarily transferred out of the trial and received F-75 coupled with additional treatment. In field practice, however, many growth faltering infants have diarrhea, may be febrile, and/or are fluid restricted for a variety of reasons (e.g. those with cardiac defects, those who are mixed fed and lack access to clean drinking water, etc.). In these situations, the higher solute load of F-100 may pose a risk.

Wilkinson's (2008) [49] approach emphasized the use of the supplemental suckling technique (SST) to facilitate relactation and lowered the amount of supplemental milk as infants recovered. However, the trial did not have sufficient resources to record further breastfeeding data or to support those who were transferred to under-resourced hospitals that lacked specialized pediatrics care and skilled lactation support. The study population of Islam et al (2020)

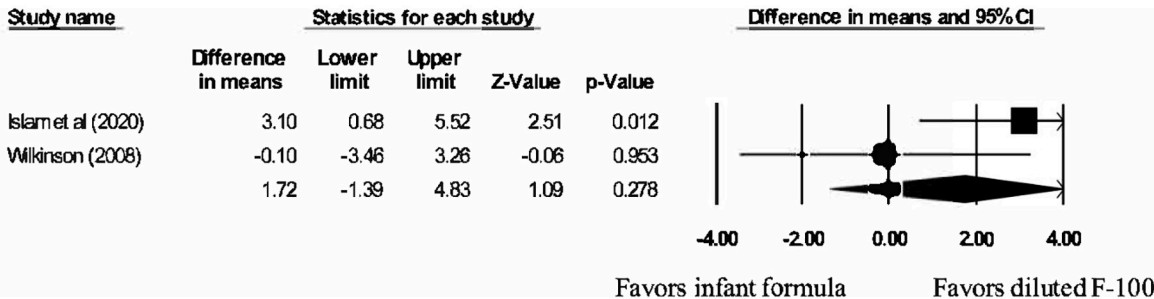

**Fig 2. Meta-analysis of weight gain.**

[48] reflects a very short period of exclusive breastfeeding and any breastfeeding at the time of trial (~50%). Although mothers who were breastfeeding received encouragement to continue and those non-breastfeeding were counseled on relactation, infants only received a very small portion of their energy intake from breastfeeding. Additionally, infants were offered water between feeds, which is not consistent with WHO global breastfeeding guidance [50].

The trials in this review demonstrate the continued need to support lactation in the context of growth faltering as part of an integrated approach protecting, promoting, and supporting breastfeeding. Although WHO guidance is clear that exclusive breastfeeding for 6 months ensures optimal infant feeding and development, to date, the focus of growth faltering interventions has been on the supplemental milks that may be used in these contexts without paying enough attention to supporting breastfeeding or relactation. A recent observational study [51] of LBW infants in resource limited settings similarly identified a need for "universal and consistent" lactation support. Moreover, a recent systematic review [52] identified additional need for high quality studies of interventions to address breastfeeding difficulties among mothers of infants under 6 months with growth faltering. Our review identified a study protocol that addresses growth faltering in infants under 6 months in Asia and Africa in the context of fully implementing WHO Breastfeeding Counselling Guidelines on breastfeeding promotion and support [53]. This study has not started recruitment.

## Limitations

Overall, there is very limited high-quality evidence on interventions to understand the nutritional management of infants under 6 months experiencing growth faltering. Of the two trials identified by this search, only one has been published in the peer reviewed literature. Both trials had small sample sizes, and high numbers of participants leaving the trial before discharge criteria were met. This limits the generalizability of the findings. In Islam et al's (2020) [48] trial the authors noted that the F-100 study arm differed between the other two study arms in greater wasting at enrollment. Additionally, over a third of infants temporarily paused their participation in the trial while transferred out due to illness, and then were later returned to the trial, which poses methodological challenges for comparability. For Wilkinson's (2008) [49] trial the inability to adequately power the trial and the lack of blinding posed challenges. There were additional non-statistically significant differences in the findings between the two centers, with longer duration of treatment from admission to discharge for the diluted F-100 group at Baraka versus longer duration for the IF group at Uvira. Finally, the design of the two studies differed substantially, and only one outcome (rate of weight gain) was shared among them for the diluted F-100 and IF comparison. The lack of studies combined with the low quality of the only two studies available is quite concerning given the strong need for

evidence-based guidelines for the nutritional management of young malnourished children. This gap has been identified for decades. It is indeed striking that many national guidelines on the treatment of acutely malnourished children, such as Afghanistan [54], Ethiopia [55], and Myanmar [56], include F-75, diluted F-100 or infant formula for infants under 6 months in the recommendations without having the level of evidence needed to support this.

Another key concern in this area is the lack of a well-thought-out framework that can help improve the design for studies in this area, which accounts for the need to first and foremost protect breastfeeding among malnourished infants. A current Cochrane review is examining whether lower protein content formulas can address malnutrition without the adverse longer-term outcomes of obesity and metabolic disorders [57]. High income settings face additional considerations on the use of fortifiers and modular nutrients for preterm infants, including cost and cross-contamination of using pumps to express milk to nutrient enrich a supply [58].

## Towards a new research framework

We specifically propose a new research framework that is consistent with the MAMI pathway [59] package's emphasis on continuing/restoring breastfeeding whenever possible, and supports continuation of breastfeeding, relactation counseling, and expands this framework with the use of human milk. We recommend conducting a series of linked studies to inform an evidence-based decision tree to understand if, when and for how long the malnourished child needs to be supplemented with supplemental milks and which kind of supplemental milks are needed (Fig 3). In full consistency with our systematic review's conclusion, the revised 2023 WHO guideline on the prevention and management of wasting and nutritional oedema (acute malnutrition) in infants and children under 5 years also emphasizes the importance of

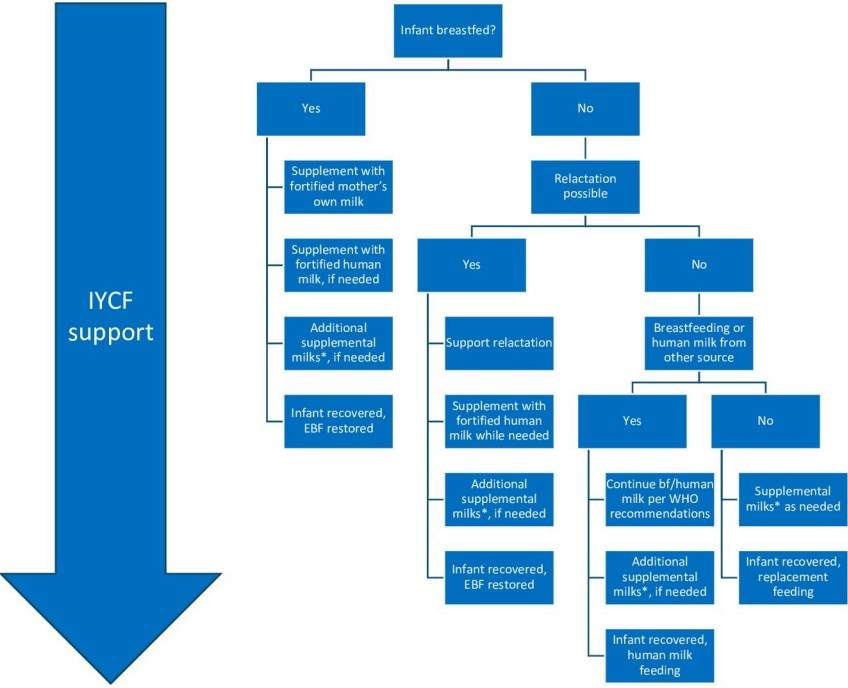

**Fig 3. Research framework to guide the design of future studies to inform comprehensive evidence-based guidelines for recovery from growth faltering in infants <6 months.** Note: Supplemental milk is defined as donor human milk, fortified donor human milk, fortified mothers' own milk, human milk from wet nurse, commercial infant formula, nutrient-enriched commercial infant formula, F-75, F-100, or diluted F-100.

supporting and restoring optimal breastfeeding practices to prevent malnutrition and in the context of continued support for recovery from acute malnutrition [60]. In caring for hospitalized infants that are at elevated risk of growth faltering (e.g. very low birth weight infant or infants with cardiac defects), there has been a paradigm shift in the past 30 years from the almost exclusive use of nutrient-enriched specialty formulas to a variety of strategies to preserve human milk feeding, but without rigorously designed research studies. These strategies have included breastfeeding support, increasing the volumes of breast milk fed, and adding nutrient fortifiers or less expensive nutrient modulars to expressed human milk (mother's own or donor human milk) fed by feeding bottle, tube or a supplemental nursing system. 2022 WHO guidance on caring for preterm and LBW infants states that mother's own milk (MOM) is recommended for this group, and there is limited to no benefits to using IF in comparison to MOM [6]. The evidence base for the guidance suggests that breastfeeding is likely an undervalued intervention for these vulnerable infants. Although this guidance is not specifically directed at infants with growth faltering, it has implications for interventions to address growth faltering in settings with high prevalence of malnutrition. Given that practitioners need to make feeding decisions with very little evidence to guide them, it is important to also learn from successful practical experiences managing the nutritional needs of infants who are not growing well across diverse settings, and carefully evaluate them.

The trials also call attention to the importance of translation of research through implementation of interventions that is responsive to the needs of diverse local contexts with a high prevalence of growth faltering. A key issue for both trials in our analysis was that there were infants leaving the trial before discharge criteria were met. When caregivers have competing demands on their time, they may struggle to keep their infants in treatment, and sustaining recovery may also be a challenge. They may have labor and other caregiving demands that prevent them from being able to stay at a hospital or recovery center. To ensure recovery and sustained growth and healthy development, a broader evidence-informed policy approach is necessary to ensure support for families facing these challenges.

## Conclusion and research recommendations

Critical questions remain about how to best prevent, and support recovery, from growth faltering in the context of malnutrition. The two studies we identified in this systematic review provide limited generalizability. The inequities in the prevalence of growth faltering under 6 months between those in low- and middle-income settings compared with high-income settings point to the importance of addressing the broader social inequities that are the upstream drivers of malnutrition [61]. Additional research needs to address the needs of infants at risk for growth faltering and those facing complex medical conditions. We recommend a paradigm shift for this research that reorients breastfeeding and the use of human milk and skin to skin contact as foundational even to medically fragile infants. In this paradigm, the focus in malnutrition research would be on supporting and restoring breastfeeding, rather than previous approaches that focus research questions on supplemental milk products. A similar paradigm shift has already taken place in preterm infant feeding research and practice, as reflected by the 2022 WHO recommendations for care of the preterm or low-birth-weight infant [6]. Further investment is necessary to ensure that the protection, promotion and support of breastfeeding is at the center of this work, even in complex humanitarian emergencies.

## Supporting information

**S1 Checklist. PRISMA 2020 checklist.**
(DOCX)

**S1 Appendix. Search strategy.**
(PDF)

**S2 Appendix. Excluded studies after discussion.**
(PDF)

## Acknowledgments

The authors would like to thank the Johns Hopkins University Welch Library team for their assistance.

## Author Contributions

**Conceptualization:** Cecília Tomori, Deborah L. O'Connor, Mija Ververs, Rafael Pérez-Escamilla.

**Data curation:** Dania Orta-Aleman, Katerina Paone.

**Formal analysis:** Cecília Tomori, Dania Orta-Aleman, Katerina Paone, Chakra Budhathoki, Rafael Pérez-Escamilla.

**Funding acquisition:** Cecília Tomori.

**Investigation:** Cecília Tomori, Deborah L. O'Connor, Mija Ververs, Dania Orta-Aleman, Katerina Paone, Chakra Budhathoki, Rafael Pérez-Escamilla.

**Methodology:** Cecília Tomori, Deborah L. O'Connor, Mija Ververs, Dania Orta-Aleman, Katerina Paone, Chakra Budhathoki, Rafael Pérez-Escamilla.

**Project administration:** Cecília Tomori, Dania Orta-Aleman, Katerina Paone.

**Resources:** Cecília Tomori.

**Software:** Cecília Tomori.

**Supervision:** Cecília Tomori, Dania Orta-Aleman, Rafael Pérez-Escamilla.

**Validation:** Cecília Tomori, Dania Orta-Aleman, Katerina Paone.

**Visualization:** Dania Orta-Aleman, Chakra Budhathoki.

**Writing – original draft:** Cecília Tomori, Rafael Pérez-Escamilla.

**Writing – review & editing:** Cecília Tomori, Deborah L. O'Connor, Mija Ververs, Dania Orta-Aleman, Katerina Paone, Chakra Budhathoki, Rafael Pérez-Escamilla.

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
