## [Decision Letter · Decision Letter 0]

28 Jun 2023

PGPH-D-23-00601

Critical Research Gaps in Treating Growth Faltering in Infants under 6 Months: A Systematic Review and Meta-Analysis

Dear Dr. Tomori,

Thank you for submitting your manuscript to PLOS Global Public Health. After careful consideration, we feel that it has merit but does not fully meet PLOS Global Public Health’s publication criteria as it currently stands. Therefore, we invite you to submit a revised version of the manuscript that addresses the points raised during the review process.

We look forward to receiving your revised manuscript.

Kind regards,

Tinku Thomas, Ph.D

Academic Editor

Journal Requirements:

2. We have noticed that you have uploaded Supporting Information files, but you have not included a list of legends. Please add a full list of legends for your Supporting Information files after the references list. 

Additional Editor Comments (if provided):

Reviewers' comments:

Reviewer's Responses to Questions

**Comments to the Author**

1. Does this manuscript meet PLOS Global Public Health’s publication criteria? Is the manuscript technically sound, and do the data support the conclusions? The manuscript must describe methodologically and ethically rigorous research with conclusions that are appropriately drawn based on the data presented.

Reviewer #1: Yes

Reviewer #2: Yes

Reviewer #3: Yes

2. Has the statistical analysis been performed appropriately and rigorously?

Reviewer #1: Yes

Reviewer #2: Yes

Reviewer #3: Yes

3. Have the authors made all data underlying the findings in their manuscript fully available (please refer to the Data Availability Statement at the start of the manuscript PDF file)?

Reviewer #1: Yes

Reviewer #2: Yes

Reviewer #3: Yes

4. Is the manuscript presented in an intelligible fashion and written in standard English?

Reviewer #1: Yes

Reviewer #2: Yes

Reviewer #3: Yes

5. Review Comments to the Author

Reviewer #1: 1. Please define some of the statistical symbols for the reader.

2.Indicate the period of search? Is it from 1980 to when?

3. The final selection of 2 studies under systematic review:

This looks too few to make conclusive recommendations. Is it possible to expand the selection criteria? It will be nice to review studies from more developed, developing (Bangladesh not sufficient) and even Sub-Saharan Africa(DRC, Niger: not sufficient)!. Should we assume that there are only 2 studies relevant for the study? What was the main limitations in arriving at a substantial number of studies?

4. Demographic and Health Survey (DHS) are conducted in almost all regions of the world and the data is regularly available. It has a module on infant feeding and nutrition outcomes including growth faltering. Please include some findings and how they relate with current study.

5.There are recent studies on growth faltering. Consider reviewing many of these studies and what extent do these inform your study?

Reviewer #2: This article is well written, I have only two recommendations:

1. Some of the content of conclusion can be better placed in discussion section.

2. All journal titles should be abbreviated as per the ‘NLM Catalog of Journals’.

Reviewer #3: This is a well-written manuscript that only needs to undergo a few minor changes. It has clear research questions, good flow of search strategy ,and supported with evidence on excluded studies with justification. It is technically sound. Though the study has heterogeneity, the magnitude is also estimated statistically. I really enjoyed reviewing.

6. PLOS authors have the option to publish the peer review history of their article (what does this mean?). If published, this will include your full peer review and any attached files.

**Do you want your identity to be public for this peer review?** For information about this choice, including consent withdrawal, please see our Privacy Policy.

Reviewer #1: **Yes: **Gilbert Habaasa

Reviewer #2: **Yes: **Syeda Mah-e-jabeen Zehra

Reviewer #3: No

---

## [Decision Letter · Decision Letter 1]

3 Oct 2023

PGPH-D-23-00601R1

Critical Research Gaps in Treating Growth Faltering in Infants under 6 Months: A Systematic Review and Meta-Analysis

Dear Dr. Tomori,

Thank you for submitting your manuscript to PLOS Global Public Health. After careful consideration, we feel that it has merit but does not fully meet PLOS Global Public Health’s publication criteria as it currently stands. Therefore, we invite you to submit a revised version of the manuscript that addresses the points raised during the review process.

We look forward to receiving your revised manuscript.

Kind regards,

Tinku Thomas, Ph.D

Academic Editor

Journal Requirements:

Additional Editor Comments (if provided):

Reviewers' comments:

Reviewer's Responses to Questions

**Comments to the Author**

1. If the authors have adequately addressed your comments raised in a previous round of review and you feel that this manuscript is now acceptable for publication, you may indicate that here to bypass the “Comments to the Author” section, enter your conflict of interest statement in the “Confidential to Editor” section, and submit your "Accept" recommendation.

Reviewer #1: (No Response)

Reviewer #3: All comments have been addressed

2. Does this manuscript meet PLOS Global Public Health’s publication criteria? Is the manuscript technically sound, and do the data support the conclusions? The manuscript must describe methodologically and ethically rigorous research with conclusions that are appropriately drawn based on the data presented.

Reviewer #1: Partly

Reviewer #3: Yes

3. Has the statistical analysis been performed appropriately and rigorously?

Reviewer #1: Yes

Reviewer #3: Yes

4. Have the authors made all data underlying the findings in their manuscript fully available (please refer to the Data Availability Statement at the start of the manuscript PDF file)?

Reviewer #1: Yes

Reviewer #3: Yes

5. Is the manuscript presented in an intelligible fashion and written in standard English?

Reviewer #1: Yes

Reviewer #3: Yes

6. Review Comments to the Author

Reviewer #1: 1. Is this section of included studies part of the study findings or it is supposed to be under methodology? There is need for consistent flow.

2.Please present results based on the specific objectives of the study. To me, the results are not clear. They are more of methodological explanations rather than reporting was found out in line with the study objectives. Short of this, the reader gets lost because results look like extended sections of methodology section.

3.Actually, the results writeup should be more than the methodology section.

4.Organise your manuscript according to the journal guidelines. In most cases, results are followed by discussion of results, Conclusion and recommendations.

5. Create a subheading that clearly shows the limitations of the study.

6. This table is not readable. Please use a clear table and if not readable, why include it?

7. See additional comments in the attached copy.

Reviewer #3: The manuscript has been revised well. I hope this manuscript will be

acceptable.

7. PLOS authors have the option to publish the peer review history of their article (what does this mean?). If published, this will include your full peer review and any attached files.

**Do you want your identity to be public for this peer review?** For information about this choice, including consent withdrawal, please see our Privacy Policy.

Reviewer #1: **Yes: **Gilbert Habaasa

Reviewer #3: **Yes: **Gutu Yonas Kitesa

---

## [Editor Report · Decision Letter 2]

15 Nov 2023

Critical Research Gaps in Treating Growth Faltering in Infants under 6 Months: A Systematic Review and Meta-Analysis

PGPH-D-23-00601R2

Dear Dr. Tomori,

We are pleased to inform you that your manuscript 'Critical Research Gaps in Treating Growth Faltering in Infants under 6 Months: A Systematic Review and Meta-Analysis' has been provisionally accepted for publication in PLOS Global Public Health.

Best regards,

Tinku Thomas, Ph.D

Academic Editor